# Molecular Characterization and Antibacterial Resistance Determination of *Escherichia coli* Isolated from Fresh Raw Mussels and Ready-to-Eat Stuffed Mussels: A Major Public Health Concern

**DOI:** 10.3390/pathogens13070532

**Published:** 2024-06-24

**Authors:** Artun Yibar, Izzet B. Saticioglu, Nihed Ajmi, Muhammed Duman

**Affiliations:** 1Department of Food Hygiene and Technology, Faculty of Veterinary Medicine, Bursa Uludag University, Bursa 16059, Turkey; artunyibar@uludag.edu.tr; 2Department of Aquatic Animal Disease, Faculty of Veterinary Medicine, Bursa Uludag University, Bursa 16059, Turkey; iburcinsat@gmail.com (I.B.S.); nihed.ajmi.95@gmail.com (N.A.)

**Keywords:** *Escherichia coli*, mussel, antibacterial resistance, phylogenetic analysis, food safety, public health

## Abstract

Our study focused exclusively on analyzing *Escherichia coli* (*E. coli*) contamination in fresh raw mussels and ready-to-eat (RTE) stuffed mussels obtained from authorized and regulated facilities. However, it is critical to recognize that such contamination represents a significant public health threat in regions where unauthorized harvesting and sales practices are prevalent. This study aimed to comprehensively assess the prevalence, molecular characteristics, and antibacterial resistance profiles of *E. coli* in fresh raw mussels and RTE stuffed mussels. *E. coli* counts in fresh raw mussel samples ranged from 1 to 2.89 log CFU/g before cooking, with a significant reduction observed post-cooking. RTE stuffed mussel samples predominantly exhibited negligible *E. coli* presence (<1 log CFU/g). A phylogenetic analysis revealed a dominance of phylogroup A, with variations in the distribution observed across different sampling months. Antibacterial resistance was prevalent among the *E. coli* isolates, notably showing resistance to ampicillin, streptomycin, and cefotaxime. Extended-spectrum β-lactamase (*ESβL*) production was rare, with only one positive isolate detected. A variety of antibacterial resistance genes, including *tetB* and *sul1*, were identified among the isolates. Notably, virulence factor genes associated with pathogenicity were absent. In light of these findings, it is imperative to maintain rigorous compliance with quality and safety standards at all stages of the mussel production process, encompassing harvesting, processing, cooking, and consumption. Continuous monitoring, implementation of rigorous hygiene protocols, and responsible antibacterial drug use are crucial measures in mitigating food safety risks and combating antibacterial resistance. Stakeholders, including seafood industry players, regulatory agencies, and healthcare professionals, are essential to ensure effective risk mitigation and safeguard public health in the context of seafood consumption.

## 1. Introduction

Mussels are a nutritious shellfish that are rich in protein, minerals, vitamins, and omega-3 fatty acids. They can be consumed both cooked and fresh. They are also low in fat (less saturated fat) and calories, making them a healthy choice for a balanced diet. The consumption of mussels can provide various health benefits, such as improving immune function, reducing inflammation, enhancing brain function, and preventing anemia [1,2]. Mussels are also considered a sustainable and eco-friendly seafood option, as they do not require feed or fertilizers and can filter and improve water quality [1].

Stuffed mussels are a popular traditional food sold by vendors in the Mediterranean Sea countries. The mussel variety used for stuffed mussels is *Mytilus galloprovincialis*, which is commonly known as the “black mussel”. Ready-to-eat stuffed (RTE) mussels are typically prepared by first cleaning the cockleshells thoroughly, removing all feather-like structures, and then stuffing them with a mixture of rice, oil, salt, and spices. The cockleshells are then closed firmly and steamed [3].

*E. coli* is widely recognized as a key indicator of fecal contamination and is used to assess the microbiological quality of seafood, particularly mussels. Its presence in food, especially those that are RTE, can signal potential contamination with pathogenic microorganisms. *E. coli* can cause severe illnesses in humans, including gastroenteritis, urinary tract infections, and neonatal meningitis. Some strains, like EHEC, produce Shiga toxins that can lead to severe conditions such as hemorrhagic colitis (HC) and hemolytic uremic syndrome (HUS), which is a leading cause of acute kidney failure in children. Additionally, some of other serotypes of *E. coli* possess higher risks, such as carrying the stx and eae virulence genes, named STEC *E. coli*. These infections, often associated with *E. coli*, can result from consuming contaminated and undercooked foods and may also be transmitted through the fecal–oral route [4]. It can contaminate marine environments and mussels through various sources like sewage, agricultural runoff, and animal manure and consequently enter the food chain [5,6]. The prevalence of *E. coli* serotypes in both the marine environment and mussels is influenced by factors such as environmental conditions, seasonal changes, mussel species, and pollution levels in harvesting areas [7,8,9]. These serotypes may possess different surface antigens affecting their pathogenicity and host range. The impact of these factors on public health is significant, especially for vulnerable populations [10,11]. *E. coli* species may also harbor antibacterial resistance genes, posing a global concern [12].

Despite the nutritional benefits of mussels, harvesting areas that are polluted, with improper preparation, storage, and sale conditions, can lead to *E. coli* contamination. Contaminated water, exposure to temperature fluctuations, and processing with contaminated equipment can result in cross-contamination and rapid bacterial growth. The consumption of raw or undercooked mussel products may lead to serious illnesses. Consequently, the presence of *E. coli* in seafood products, particularly mussels, requires a thorough risk assessment to address potential health risks to consumers [13,14]. Numerous risk assessment studies have focused on mussels and mussel products to ensure food safety [15,16,17,18,19].

While mussels and mussel products are widely consumed, there is a significant deficiency in evaluating and characterizing *stx* and *eae*-carrying *E. coli* isolates found in fresh raw mussels and RTE stuffed mussels, as well as along the RTE stuffed mussel processing line. A molecular characterization of *E. coli* serotypes and their AMR genes is of vital importance for the control of the spread of these genes and factors and for the implementation of appropriate prevention and treatment strategies.

The objective of this study was to identify the primary source of contamination of *E. coli* in fresh raw mussels and RTE stuffed mussels. Additionally, this study aimed to detect the presence of *stx* and *eae*, as well as other virulence factors and AMR genes, in a survey of fresh raw mussels and RTE stuffed mussels collected from four different companies in four different regions of Turkey.

## 2. Materials and Methods

### 2.1. Sample Collection

During the period between June 2022 and May 2023, RTE stuffed mussels (n = 25) and fresh raw mussels (n = 25) used in the production of these stuffed mussel samples were collected regularly every month (all samples were taken on the same day and were considered as a single batch) from three different companies harvesting and processing mussels from the Marmara Sea, which is an inner sea surrounded by heavily populated cities and industrial areas. All the selected companies applied the Hazard Analysis and Critical Control Points (HACCP) and Good Manufacturing Practices (GMP) systems in their production. The locations of harvesting for these companies were as follows: R1: Balikesir (40°34′41.8″ N 27°35′37.8″ E), R2: Mudanya (40°22′59.8″ N 28°52′45.3″ E), and R3: Gemlik (40°28′18.2″ N 28°54′28.3″ E). In addition, another company (R4: Istanbul [40°59′31.1″ N 29°00′46.1″ E]) was included in this study to investigate point-by-point possible *E. coli* contamination points. A total of nine sampling points (SP) were selected as follows: SP1: raw mussel, SP2: swab from knives, SP3: shelling step, SP4: swab from handlers’ hands, SP5: stuffing with pre-cooked rice and spices, SP6: cooking with an aromatic blend of rice and spices, SP7: portioning and packaging, SP8: shipment, SP9: ready-to-eat stuffed mussel at selling point. Sterile swabs were used to sample the food handlers’ hands (after routine cleaning procedures) and the knives before they were engaged in food preparation. Swabbing from both the handlers’ hands and the knives was performed on a 10 cm^2^ area. All the samples were packed in sterile bags and transferred to the laboratory in cold chain within three hours. The cooking periods applied at R4 were 65 °C for 17.6 ± 1.8 min for precooking (rice and spices) and 72 °C for 20.6 ± 0.9 min for main cooking (mussels, rice, and spices together). Detailed information about the regions, sampling points, and processes are provided in Appendix A.

### 2.2. Isolation and Enumeration of E. coli

A total of 25 samples were collected on a single day and considered a single batch. The inter-shell contents of each batch were mixed, and 10 g of the mixture from each batch was transferred to sterile stomacher bags (Seward Medical, London, UK). Then, 90 mL of sterile Maximum Recovery Diluent (MRD, Oxoid, ThermoFisher, Milano, Italy) was added and homogenized. Subsequently, 10-fold serial dilutions were prepared from the MRD. *E. coli* isolation and enumeration were conducted using the pour plate method on Tryptone Bile X-Glucuronide Agar (TBX, Oxoid, Basingstoke, UK). Following inoculation from 1 mL of the appropriate dilutions, the plates were incubated at 37 ± 2 °C for 4 h and then at 44 °C for 20 h. Colonies with a blue–green color were identified and enumerated as log colony-forming units per gram (log CFU/g) [7,20]. Additionally, suspected *E. coli* colonies were isolated using the Violet Red Bile Agar (VRB, Oxoid, Hampshire, UK) double-layer pour plate method. The inoculated plates were incubated at 37 °C for 24 h. Following incubation, purple colonies (pinkish red colonies with bile precipitate) were considered to be *E. coli* [21]. The blue–green colonies isolated from the TBX agar and the suspected colonies from the VRB agar were subjected to subculturing in Tryptic Soya Broth (TSB, Oxoid, Thermofisher, Madrid, Spain) and on Tryptic Soya Agar (TSA, Oxoid, Thermofisher, Madrid, Spain) at 37 °C for 24 h. Thereafter, all *E. coli* isolates were confirmed using standard biochemical tests.

### 2.3. Identification of E. coli Isolates

The confirmation of *E. coli* isolates was achieved using polymerase chain reaction (PCR) amplification of the *E. coli*-specific universal stress protein (*uspA*) and *uidA* genes following the procedure outlined by Chen and Griffiths (1998) [22] and Heijnen and Medema (2006) [23], respectively (Table 1).

### 2.4. Determination of Phylogenetic Groups of E. coli Isolates 

*E. coli* isolates were classified into phylogenetic groups (A, B1, B2, and D) in accordance with the Clermont’s method (2000 and 2013) [24,25], which involved the identification of two virulence genes, *chuA* (encoding a hem transporter protein in *E. coli* O157: H7), *yjaA* (encoding a hypothetical protein initially identified in the genome of *E. coli* K-12), and one DNA fragment (*TspE4.C2*) and *arpa* (ankyrin repeat protein) genes [26] (Table 1).

### 2.5. Antibacterial Resistance Testing of E. coli Isolates

The antibacterial resistance of *E. coli* isolates was evaluated using the Kirby–Bauer disc diffusion method in accordance with the Clinical and Laboratory Standards Institute (CLSI) guidelines [25]. A set of antibacterials (Oxoid, Basingstoke, UK) was utilized, including tetracycline (TE; 30 µg), chloramphenicol (CL; 30 µg), trimethoprim-sulfamethoxazole (SXT; 25 µg), streptomycin (STR; 10 µg), ciprofloxacin (CIP; 5 µg), levofloxacin (LEV; 5 µg), and ampicillin (AM; 10 µg). The isolates’ resistance was evaluated according to the CLSI guidance (2023) [27], with *E. coli* ATCC 25922 used as a quality control strain, and classified as sensitive, intermediate resistant, or resistant.

### 2.6. Phenotypic Test for the Presence of Extended-Spectrum β-Lactamase (ESβL) in E. coli Isolates 

The presence of Extended-Spectrum β-Lactamase (*ESβL*) in *E. coli* isolates was also evaluated. Initially, the CLSI-guided (2023) [27] *ESβL* test was conducted using a cefotaxime (CTX; 30 µg) antibacterial disc, with the isolates classified as sensitive, intermediate, or resistant. The confirmation test was performed according to the guidelines of the European Committee on Antimicrobial Susceptibility Testing (EUCAST, 2023) [28] using the double-disc synergy test (DDST) with amoxicillin-clavulanic acid (AMC; 20/10 μg), ceftazidime (CAZ; 30 μg), cefepime (FEP; 30 μg), and cefotaxime (CTX; 30 μg). The presence of a synergistic effect, demonstrated by a clear inhibition zone, indicates *ESβL* production. The CLSI methods (2023) [27] were employed, with cefotaxime (CTX; 30 µg) and ceftazidime (CAZ; 30 µg) discs plated both individually and in combination with clavulanic acid (CLA; 10 µg) on a Mueller Hinton Agar (MHA, Oxoid, Basingstoke, UK) plate to confirm *ESβL*. *ESβL*-positive phenotypes were confirmed by a ≥5 mm enhancement in the zone of inhibition in CTX-CLA or CAZ-CLA compared to the respective antibacterial disc alone (combined disc test, CDT). The enlarged inhibition zone indicated the isolates’ ability to neutralize clavulanic acid.

### 2.7. Identification of β-Lactamase and Antibacterial Resistance Genes of E. coli Isolates

The investigation of major beta-lactamase genes, namely *bla_CTX-M_*, *bla_TEM_*, *bla_SHV_*, and *bla_OXA_* within the *E. coli* isolates was conducted via PCR in accordance with the methodology outlined by Ogutu et al. (2015) [29]. Furthermore, an analysis was conducted on the isolates to determine the presence of genes associated with resistance to various antibacterial agents, including tetracycline (*tetA* and *tetB*), sulfonamides (*sul1*, *sul2*, and *sul3*), florfenicol/chloramphenicol (*floR*), and quinolones (*qnrA* and *qnrB*) [30,31].

### 2.8. Virulence Factor Genes of E. coli Isolates

*E. coli* isolates were tested using conventional PCR for the following virulence factors: Shiga-like toxin genes (*stx1* and *stx2*), bundle-forming pilus (*bfpA*), attaching and effacing factor gene (*eae*), O157 antigen (*rfbE*), flagellar antigen (*flic*), O26 and O103 O-antigen (*wzx*), O145 (*ihp1*), and O111 (*wbdl*) genes, with primers and conditions outlined by ISO (2012) [32] and Clermont et al. (2013) [25].

## 3. Results

### 3.1. Isolation, Enumeration, and Identification 

The *E. coli* counts prior to the cooking process ranged from 1 to 2.89 log CFU/g. The mean *E. coli* counts in the fresh raw mussel samples from R1, R2, R3, and R4 were 1.10, 1.96, 1.86, and 1.72 log CFU/g, respectively. The highest *E. coli* count was detected in the fresh raw mussels from R3 in September, followed by 2.83 log CFU/g in the fresh raw mussels from R4. Seasonally, the period with the least contamination was spring, with growth observed only in the fresh raw mussels from R1 in March. No *E. coli* was detected in any samples during April and May. The highest counts were obtained from samples collected in the autumn period. In December, a single colony from VRB tested positive for *E. coli* in RTE stuffed mussels from R3.

*E. coli* was not isolated at any stage after the cooking process (SP6-SP9 in R4). The counts of *E. coli* were found to be 1.85 log CFU/g before the cooking process and <1 log CFU/g in the RTE stuffed mussel samples by the destruction of 0.90 log after the cooking process. The RTE stuffed mussel samples contained no *E. coli*, and only one isolate from VRB was *E. coli* positive (60EM, from R3). No *E. coli* was detected in any sample from the food contact surfaces (SP2 and SP4) of the RTE stuffed mussel production line (Figure 1).

The genetic diversity of the isolates we obtained as a result of isolation and PCR identification, which we have provided in Appendix A, were investigated. To this end, a total of 74 isolates were collected: 44 from fresh raw mussels, 15 from SP3 (shelling step), 14 from SP5 (stuffing with pre-cooked rice and spices), and one from an RTE stuffed mussel sample.

From a seasonal perspective, the highest number of *E. coli* isolates was obtained from samples in October (16 isolates), followed by September (15 isolates). These isolates, used for phenotypic antibacterial resistance and genetic diversity research, are referred to by a “number and EM” code throughout the remainder of this study.

### 3.2. Determination of Phylogenetic Groups 

Two different groupings were made for the phylogenetic analysis of the *E. coli* isolates (n = 74) obtained in our study. Phylogroup A was the most dominant (38/74; 51.4%) phylogroup among the isolates, followed by phylogroups B1 (18/74; 24.3%), B2 (13/74; 17.6%), and D (5/74; 6.8%) based on Clermont et al. (2000) [24]. All phylogroup B1 *E. coli* isolates harbored the *arpA* gene. All the samples isolated in January and February 2023 (excluding 56EM) belonged to phylogroup A1. In the other months, the isolates showed a heterogeneous distribution (Appendix A). In the phylogenetic grouping of *E. coli* isolates, according to Clermont et al. (2013) [25], the most common phylogroup was A (16/74; 21.6%), followed by A or C (11/74; 14.9%), B1 (10/74; 13.5%), B2 (5/74; 6.8%), Clade1 or 2 (2/74; 2.7%), E or Clade1 (2/74; 2.7%), D or E (1/74; 1.4%), and F (1/74; 1.4%). Just over 35.1% (26/74) of the isolates were non-typable (Appendix A).

### 3.3. Phenotypic and Genotypic Identification of Antibacterial Resistance and ESBL of E. coli Isolates

The results of the isolates’ phenotypic resistance to antibacterials are provided in Appendix A. The intermediate resistant and resistant *E. coli* phenotypes were as follows: ampicillin (35 isolates; 47.3%), streptomycin (21 isolates; 28.4%), cefotaxime (15 isolates; 20.3%), trimethoprim-sulfamethoxazole (11 isolates; 14.9%), tetracycline (10 isolates; 13.5%), chloramphenicol (six isolates; 8.1%); ciprofloxacin (five isolates; 6.8%), and levofloxacin (three isolates; 4.1%). Out of the 74 isolates, 25 isolates (33.8%) exhibited no resistance to any antibacterials. Only one isolate (41EM) isolated in October was *ESBL* positive. A majority of the *E. coli* isolates demonstrated high susceptibility rates to levofloxacin and ciprofloxacin, with over 95% of the isolates remaining sensitive.

Cephalothin-*bla_SHV_* was not detected in any *E. coli* isolates. *tetB* (10/74; 13.5%), *sul1* (3/74; 4.1%), *sul2* (9/74; 12.2%), *sul3* (2/74; 2.7%), *qnrA* (1/74; 1.4%), *qnrB* (1/74; 1.4%), and *floR* (1/74; 1.4%) were detected (Appendix A).

### 3.4. Identification of Virulence Factor Genes of E. coli Isolates

The Shiga-like toxin (*stx1* and *stx2*), attaching and effacing factor (*eae*), bundle-forming pilus structural gen (*bfp*A), O157 antigen (*rfbE*), O26 and O103 O-antigen (*wzx*), and O145 (*ihp1*) genes were not detected. *wbdl* (O111 gene) was detected in only one isolate (23EM). *flicH7* (flagellar antigen) was found in three isolates (7EM and 10EM [June and August, respectively] and 12EM [September]).

## 4. Discussion

*E. coli* species, which serve as an indicator of fecal contamination and are consequently an important food safety indicator, can reach prevalences of up to 30% in shellfish [33]. In India, Singh et al. (2020) [34] isolated a total of 150 *E. coli* isolates from four different types of fresh shellfish. Our results showed that *E. coli* was enumerated in 35.4% (17/48) of fresh raw samples and none of the stuffed mussels, with a mean count of 2.15 log CFU/g. Raw mussels can harbor many important pathogens such as *E. coli* through municipal sewage discharge, industrial wastewater loads, rainfall or irrigation water runoff over land, and the release of contaminants into streams, lakes, or coastal waters [17].

In the current study, a comprehensive monthly sampling approach allowed for monitoring and ensuring the mussels’ microbiological safety and quality throughout their preparation, handling, and distribution processes. The absence of *E. coli* in the hand and knife swab samples showed that the food handlers complied with hygiene and sanitation rules. A study conducted in Istanbul Province in Turkey examined the microbiological quality of RTE stuffed mussels according to the Turkish Food Codex. The results revealed that 77% of the samples had coliform bacteria and 22% had *E. coli* [35]. In another study conducted in Ankara Province in Turkey, 30% of the analyzed stuffed mussel samples were not suitable for consumption due to the presence of *E. coli* [13]. Unlike other studies, which detected very high levels of *E. coli* contamination in RTE stuffed mussel, in our study, only one (2.1%) *E. coli* isolate was identified in the RTE stuffed mussel samples. This can be due to the inactivation of the bacteria during heating processes, the prevention of contamination, or any other difference in the processing steps. *E. coli* contamination in RTE stuffed mussels can be significantly reduced under stringent processing conditions. Strict hygiene protocols should be followed, including regular hand washing, wearing gloves, and using sanitized equipment by all food handlers involved in the processing. The processing environment should be controlled, with limited access to prevent external contamination, and with regular cleaning and disinfection conducted. All equipment and utensils used in the processing should be sterilized before use. Mussel products should be cooked at temperatures exceeding 70 °C to ensure the inactivation of *E. coli* and other potential pathogens. Additionally, the final product should be sold under appropriate conditions to prevent post-processing contamination. It is also important to consider that the levels of contamination could vary due to changes in pollution rates in the marine environments where the mussels are harvested. Implementing these comprehensive measures can help ensure the safety of RTE stuffed mussels [17,36,37].

Bazzoni et al. (2019) [38] reported that the presence of *E. coli* in raw mollusk samples was more pronounced in the fall and winter seasons (270 and 330 MPN/100 g, respectively). They noted that they did not encounter contamination during the summer season. Similarly, Sferlazzo et al. (2018) [39] mentioned that *E. coli* contamination tended to be lower during the summer season. In a study conducted in Italy, *E. coli* was detected in all samples of *M. galloprovincialis* and *Ruditapes decussatus* [40]. In another study conducted in Italy, which examined 600 raw mussels, *E. coli* was detected in 3.5% of the samples [41]. Notably, *E. coli* was not detected during these months, indicating either successful microbial control measures or seasonal conditions not conducive to *E. coli* survival.

Phylogenetic groups of *E. coli* have also been associated with different ecological niches, virulence factors, and antibacterial resistance patterns [42,43]. Therefore, the presence and diversity of *E. coli* phylogenetic groups in mussels may vary depending on the origin, season, and treatment of the products. In addition, PCR detection of phylogenetic groups of *E. coli* from fresh raw mussels and RTE stuffed mussels can provide useful information about the microbial quality and safety of these products. It can also help to identify the possible sources of fecal contamination and to monitor the effectiveness of processing methods. Furthermore, it can contribute to the understanding of the epidemiology and ecology of *E. coli* in aquatic environments and food chains. *E. coli* isolates categorized within phylogroup A are primarily commensal [44]. In our study, 6.8% and 1.4% of the identified isolates belonged to groups B2 and D, respectively, which are considered pathogenic. Based on the classification, 26 of the isolates belonged to an unknown phylogroup, which requires different typing methods such as MLST [23]. Phylogroups B1 and A are the most prevalent across multiple months, suggesting these are the common isolates in the environment studied. Groups B2 and D also appeared but were less frequent. The dominance of Group A and occasional appearances of Group D between December and February suggests that some isolates are more adapted to colder conditions.

The emergence of multidrug-resistant (MDR) foodborne pathogens, defined as acquired resistance to at least one antibacterial agent in three or more antibacterial categories, is considered a significant challenge in public health, with MDR *E. coli* recognized as a prominent issue in ensuring food safety [12,45,46]. In the present study, the antibacterial resistance profiles and resistance genes of the *E. coli* isolates were evaluated using disk diffusion and PCR methods. Our findings revealed that 66.2% (49/74) of the isolates exhibited resistance to at least one antibacterial, with 32.4% having resistance to two or more classes of antibacterials and thus being classified as MDR. The resistance included resistance to CTX, which is utilized as a last-resort option for treating severe infections caused by *E. coli* and other Gram-negative bacteria. The highest resistance rates were observed for AMP (47.3%), STR (28.4%), and CFX (20.3%). This resistance is particularly alarming due to the role of CTX in treating severe bacterial infections and its implications for the selection of *ESβL* producers. These antibacterials are widely used in human and veterinary medicine, and their overuse may select for resistant bacteria that can be transmitted through the food chain. The emergence and spread of resistance to these antibacterials may compromise the effectiveness of the available therapeutic options and increase the risk of treatment failure and mortality [47]. The resistance patterns exhibited seasonal fluctuations, with a notable increase in AMP and CTX resistance during the cooler months (October and November). This could be attributed to seasonal changes in antibacterial drug usage patterns in agriculture and human medicine, which often influence environmental reservoirs of resistance. Our data also revealed significant resistance to commonly used antibacterials among *E. coli* isolates, with a marked seasonality in resistance patterns. The high susceptibility to fluoroquinolones offers some therapeutic reprieve, although the emergence of *ESβL* producers and resistance to critical beta-lactam antibacterials paints a complex picture of the resistance landscape. These findings highlight the importance of tailored antibacterial stewardship and proactive public health strategies to manage and mitigate antibacterial resistance effectively.

*E. coli* species may contain many antibacterial resistance genes and may adversely affect human health. Some of these genes include *bla_TEM_*, which encodes a beta-lactamase enzyme that confers resistance to penicillin and some cephalosporins; *tetA*, which encodes a tetracycline efflux pump that confers resistance to tetracycline and doxycycline; and *sul1*, a gene that encodes a sulfonamide-resistant dihydropteroate synthase enzyme that confers resistance to sulfonamides [10,11,48]. The results of the present study revealed that 11 different resistance genes were contained in the *E. coli* isolates. The most common resistance genes were *bla_TEM_* (10/74; 13.5%), *bla_OXA_
*(5/74; 6.8%), and *bla_CTX-M_* (1/74; 1.4%) which confer resistance to beta-lactams, and *tetA* (9/74; 12.2%) and *tetB* (10/74; 13.5%), which confer resistance to tetracyclines. These genes are often located on mobile genetic elements, such as plasmids and transposons, that can facilitate their horizontal transfer among bacteria [49,50]. We also detected *ESβL* in only one isolate with the *bla_CTX-M_* gene, as well as the quinolone resistance genes *qnrA* and *qnrB* in only one isolate each. These genes confer resistance to third-generation cephalosporins and fluoroquinolones, respectively, and their presence in *E. coli* isolates from food sources is of great concern for public health. The *EsβL*-associated genes *bla_TEM_* and *bla_CTX-M_* were detected in the isolates, underscoring the presence of significant resistance mechanisms that complicate treatment options.

In this study, a comprehensive molecular analysis was conducted to assess the presence of virulence and resistance genes in *E. coli* isolates. Our findings indicate a low prevalence of virulence factors among the isolates, with significant implications for food safety and public health. Strains designated as STEC are characterized by their ability to produce Shiga toxins, which are encoded by the *stx1* and *stx2* genes. The key distinguishing factor between pathogenic and non-pathogenic *E. coli* strains lies in the presence of virulence-associated genes [51]. In this study, these virulence-associated genes were not detected in any of the isolates. The non-detection of critical virulence genes such as Shiga-like toxins (*stx1* and *stx2*), the attaching and effacing factor (*eae*), and the bundle-forming pilus structural gene (*bfpA*) in the majority of the isolates suggests a reduced potential for causing severe enteropathogenic or enterohemorrhagic infections in consumers. This absence is particularly notable, as these genes are commonly associated with severe gastrointestinal diseases including hemorrhagic colitis (HC) and hemolytic uremic syndrome (HUS). Balière et al. (2015) [52] found the presence of the *stx* gene in 35% of various shellfish samples they collected, and specifically in mussels, the presence of the *stx* gene was determined as 36.5%. Martin et al. (2019) [53] identified the virulence genes *stx1*, *stx2*, and *eae* at a lower frequency (7%) in Shiga toxin-producing *E. coli* isolates obtained from Norwegian bivalves in marine environments. This holds paramount significance for consumers of shellfish, as insufficient heat treatment during the preparation of edible shellfish species can result in foodborne infections [51]. The identification of the O111 antigen (*wbdl*) in only one isolate (23EM) and the flagellar antigen H7 (*flicH7*) in three isolates (7EM, 10EM, and 12EM) highlights the presence of specific pathogenic isolates that warrant closer attention. While the prevalence of these antigens is low, their presence indicates a potential risk for pathogenicity and necessitates continuous surveillance. The O111 and H7 antigens are particularly concerning due to their association with outbreaks and severe illness in humans [4]. The detection of these antigens, even in a small number of isolates, underscores the importance of rigorous food safety protocols in the processing of mussels. RTE mussel products pose a higher risk of contamination, emphasizing the crucial need for rigorous microbial testing and control measures in food production facilities.

Our study revealed the presence of *E. coli* isolates with diverse serotypes, phylogenetic groups, virulence factors, and AMR profiles in fresh raw mussels from the Marmara Sea in Turkey. Some of these isolates may have the potential to cause human infections and pose a challenge for the treatment of these infections. In addition, future studies could focus on comparing various contamination prevention methods to determine the most effective strategies for reducing bacterial counts in RTE mussels. Such research would provide valuable insights into optimizing processing techniques to enhance food safety.

## 5. Conclusions

In conclusion, while the prevalence of *E. coli* in RTE products appears low, the unauthorized harvesting and sale of mussels in polluted coastal areas of Turkey present significant food safety risks. To address these concerns, adherence to quality and safety standards during cultivation, thorough cleaning, proper cooking, and timely consumption or storage of mussels and stuffed mussels are crucial. Additionally, strict hygiene measures in production, continuous pathogen monitoring, and responsible use of antibacterials are essential to protect public health and prevent antibacterial resistance. Collaborative efforts among stakeholders in the seafood industry, regulatory agencies, and healthcare professionals are essential to effectively mitigate these risks and ensure the safety of seafood consumers. This study demonstrated that, in addition to phenotypic and PCR-based classification, genome-based classification and serotype determination should be included in future studies.

## Figures and Tables

**Figure 1 pathogens-13-00532-f001:**
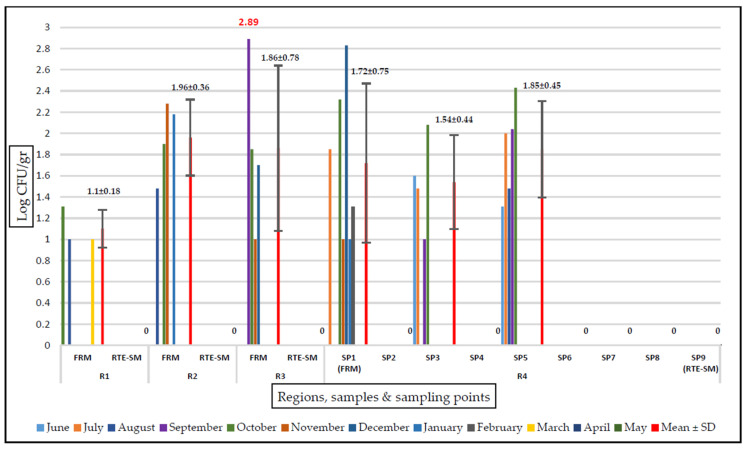
Amount of *E. coli* growth on TBX agar (log CFU/gr). FRM, fresh raw mussel; RTE-SM, ready-to-eat stuffed mussel; SP, sampling point; R, region. A value of 0 on the axis indicates that *E. coli* was not detected (< 1 log CFU/g) in this region in either the SP samples or the FRM/RTE-SM samples.

**Table 1 pathogens-13-00532-t001:** Primers used for identification and phylogenetic classification of *E. coli* isolates.

Primer ID	Target	Primer Sequence (5′-3′)	PCR Size (bp)
*uidA.f*	*uid*A	ATGGAATTTCGCCGATTTTGC	166
*uidA.r*		ATTGTTTGCCTCCCTGCTGC	
*uspA.f*	*usp*A	CCGATACGCTGCCAATCAGT	884
*uspA.r*		ACGCAGACCGTAGGCCAGAT	
*chuA.1b*	*chuA*	ATGGTACCGGACGAACCAAC	288
*chuA.2*		TGCCGCCAGTACCAAAGACA	
*yjaA.1b*	*yjaA*	CAAACGTGAAGTGTCAGGAG	211
*yjaA.2b*		AATGCGTTCCTCAACCTGTG	
*TspE4C2.1b*	*TspE4C2*	CACTATTCGTAAGGTCATCC	152
	*TspE4C2.2b*	AGTTTATCGCTGCGGGTCGC	
*AceK.f*	*arpA*	AACGCTATTCGCCAGCTTGC	400
*ArpA1.r*		TCTCCCCATACCGTACGCTA	

## Data Availability

All the data available is included in the manuscript.

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
