# Peer review of "Molecular Characterization and Antibacterial Resistance Determination of Escherichia coli Isolated from Fresh Raw Mussels and Ready-to-Eat Stuffed Mussels: A Major Public Health Concern"

_pathogens, 2024, doi:10.3390/pathogens13070532_

Round 1

Reviewer 1 Report

Comments and Suggestions for Authors

You stated that this study's objective involves raw, ready to eat stuffed mussels.  In your discussion you stated that only one E.coli strain was identified in the RTE specimens and that may be due in part to heating processes.  If the specimens are raw, what are these heating processes which makes it sound like the specimens may have been cooked.  This needs to be clarified.

You cited prevention of contamination as a reason for a reduction in bacterial counts.  It would be best to specify the methods taken to reduce contamination.  An additional study to compare different methods of reducing contamination would be useful as a followup.

Author Response

Suggestions from Reviewer #1

Authors’ response;

Dear Reviewer,

Thank you for your valuable feedback on our manuscript. We appreciate your thorough review and constructive comments, which will undoubtedly help improve the quality of our work.

Comments and Suggestions for Authors from Reviewer #1

  • You stated that this study's objective involves raw, ready to eat stuffed mussels.  In your discussion you stated that only one  coli strain was identified in the RTE specimens and that may be due in part to heating processes.  If the specimens are raw, what are these heating processes which makes it sound like the specimens may have been cooked.  This needs to be clarified.

Authors’ response: Thank you for your insightful comments and for bringing this important point to our attention. We apologize for any confusion caused by the terminology used in our manuscript.

There were two main types of products that we investigated: Raw mussels, which were directly harvested from the sea and were not processed at all, and mussels that were stuffed with rice and spices and had undergone heat processing. The term “ready-to-eat” was used only to mention those mussels which had undergone heat processing. We have revised the entire article, including the title, in order to remove any confusion.

We have also added information on our materials and methods of heat processing. We hope this clarification resolves the issue. Thank you once again for your valuable feedback.

Comments and Suggestions for Authors from Reviewer #1

  • You cited prevention of contamination as a reason for a reduction in bacterial counts.  It would be best to specify the methods taken to reduce contamination.  An additional study to compare different methods of reducing contamination would be useful as a follow-up.

Authors’ response: Regarding your comment on the statement "prevention of contamination as a reason for a reduction in bacterial counts," we apologize for not discussing the methods used to prevent contamination in our study. To address this, we have included the following details in the revised manuscript:

“E. coli contamination in RTE stuffed mussels can be significantly reduced under stringent processing conditions. Strict hygiene protocols should be followed, including regular hand washing, wearing gloves, and using sanitized equipment by all personnel involved in the processing. The processing environment should be controlled with limited access to prevent external contamination, and regular cleaning and disinfection should be conducted. All equipment and utensils used in the processing should be sterilized before use. Mussels should be cooked at temperatures exceeding 70°C to ensure the inactivation of E. coli and other potential pathogens. Additionally, the final product should be sold under appropriate conditions to prevent post-processing contamination. It is also important to consider that the levels of contamination could vary due to changes in pollution rates in the seas where the mussels are harvested. Implementing these comprehensive measures can help ensure the safety of RTE stuffed mussels.”

These methods/conditions have been described in the revised manuscript in the discussion section in Lines 279-291.

Additionally, we agree with your suggestion that a follow-up study comparing different methods of reducing contamination would be valuable. We have also included this recommendation in the Discussion section as a potential direction for future research: “In addition, future studies could focus on comparing various contamination prevention methods to determine the most effective strategies for reducing bacterial counts in RTE mussels. Such research would provide valuable insights into optimizing processing techniques to enhance food safety.” In Line 396-400.

We hope that these revisions address your concerns and improve the clarity and comprehensiveness of our manuscript. Thank you once again for your insightful comments.

Reviewer 2 Report

Comments and Suggestions for Authors

There are a number of comments on the presentation of the results:

1. It would be good to reformat Table 1 so that it does not take up so much space.

2. In table 2, it would be nice to include primers (from Chen & Griffiths 1998), which were used for species identification of E.coli.

3. It is better to present Table 3 in the form of a histogram. Then only the samples showing E.coli contamination will be visible and it will be seen in which months this happens. And the samples with the ND result will be aligned along the abscissa axis.

4. Tables 4-7 are more suitable for a grant report or database. They seem redundant in the main text of the article. The text where the clue result will be written is enough: which phylogroups, which resistances, virulent factors, etc. It would be correct to transfer the tables themselves to the supplementary.

5. I would like to see in this work data on the total contamination of samples and the proportion of E.coli in the total number of bacteria.

A general wish. In the future, it would be nice for the authors to use DNA sequencing methods for phylogenetic research, and not just PCR.

Author Response

Suggestions from Reviewer #2

Dear Reviewer,

Thank you for your valuable feedback on our manuscript. We appreciate your thorough review and constructive comments, which will undoubtedly help improve the quality of our work.

Comments and Suggestions for Authors from Reviewer #2

There are a number of comments on the presentation of the results:

  1. It would be good to reformat Table 1 so that it does not take up so much space.

Authors’ response: Thank you for your valuable feedback on our manuscript. We appreciate your suggestion to reformat Table 1 to conserve space.

We added specific points of Table 1 (formerly named) to the manuscript as text and now provide it as supplemental material (Supplementary Table S1).

Comments and Suggestions for Authors from Reviewer #2

  1. In Table 2, it would be nice to include primers (from Chen & Griffiths 1998), which were used for species identification of  coli.

Authors’ response: Thank you for your valuable feedback on our manuscript. We appreciate your suggestion to include the primers used for species identification of E. coli in Table 2 (formerly named).

In response to your suggestion, we have revised Table 1 (formerly named Table 2) to include the primers that were used for the species identification of E. coli. This addition provides a more comprehensive overview of the methods employed in our study and enhances the transparency and reproducibility of our research.

Comments and Suggestions for Authors from Reviewer #2

  1. It is better to present Table 3 in the form of a histogram. Then only the samples showing  colicontamination will be visible and it will be seen in which months this happens. And the samples with the ND result will be aligned along the abscissa axis.

Authors’ response: Thank you for your valuable feedback and insightful suggestions regarding the presentation of Table 3. We have carefully considered your recommendation to present the data in the form of a histogram.

In response to your suggestion, we have converted Table 3 into a histogram (Figure 1). This new format effectively highlights the samples showing E. coli contamination and clearly illustrates the months in which these contaminations occur. Additionally, samples with the ND (Not Detected) result as “0” are now aligned along the abscissa axis, providing a more straightforward and visually appealing representation of the data.

We believe this modification significantly enhances the clarity and impact of the presented information. The revised manuscript, including the updated figure, is attached for your review.

Comments and Suggestions for Authors from Reviewer #2

  1. Tables 4-7 are more suitable for a grant report or database. They seem redundant in the main text of the article. The text where the clue result will be written is enough: which phylogroups, which resistances, virulent factors, etc. It would be correct to transfer the tables themselves to the supplementary.

Authors’ response: Thank you for your insightful feedback regarding Tables 4-7. We have considered your suggestion carefully and agree that these tables are more suited for a supplementary section. Consequently, we have transferred Tables 4-7 to the supplementary material (Supplementary Tables S2-S5). The main text now briefly presents the key results, including the phylogroups, resistances, and virulent factors, concisely.

We appreciate your guidance in enhancing the clarity and focus of our manuscript.

Comments and Suggestions for Authors from Reviewer #2

  1. I would like to see in this work data on the total contamination of samples and the proportion of  coliin the total number of bacteria.

Authors’ response: Thank you for your thoughtful feedback and for your interest in additional data regarding the total contamination of samples and the proportion of E. coli in the total number of bacteria.

While we understand the value of including this information in the current manuscript, we regret to inform you that these specific data points are part of a broader analysis that overlaps with the scope of two other manuscripts currently under review in different journals. As a result, we are unable to incorporate these findings into this paper to avoid potential conflicts and duplication.

However, we highly appreciate your interest, and if you are interested, we would be more than willing to share these data directly with you. Please let us know if you would like us to provide this information. Thank you for your understanding and your constructive suggestions.

Within the scope of our project titled "Investigation of STEC (E. coli O157, O26, O103, O111, and O145) species and other food safety and food hygiene indicators in fresh raw mussels from the Marmara Sea and RTE stuffed mussels produced from these fresh raw mussels”, funded by the Bursa Uludag University, Scientific Research Project Association Research Grant, Project No: TOA-2022-668, we have thoroughly researched food safety indicators (total mesophilic aerobic bacteria, total coliform) and significant food pathogens (STEC species, S. aureus, Pseudomonas spp., Vibrio spp.).

We appreciate your guidance in enhancing the clarity and focus of our manuscript.

Comments and Suggestions for Authors from Reviewer #2

  1. A general wish. In the future, it would be nice for the authors to use DNA sequencing methods for phylogenetic research, and not just PCR.

Authors’ response: Thank you for your valuable feedback and suggestion regarding the use of DNA sequencing methods for future phylogenetic research. We appreciate your insight and agree that incorporating DNA sequencing methods would greatly enhance the depth and accuracy of our phylogenetic analyses. Financial strains we could not overcome prevented us from performing sequence analysis in this study. In future studies, we plan to integrate these advanced techniques alongside PCR to provide a more comprehensive understanding of the genetic relationships within our samples.

In addition, this situation is also presented as a recommendation in the discussion (line 314) and conclusion section (lines 411-413).

We hope that these revisions address your concerns and improve the clarity and comprehensiveness of our manuscript. Thank you once again for your insightful comments.

Reviewer 3 Report

Comments and Suggestions for Authors

This manuscript had a basic investigation on the prevalence, molecular characteristics, and antibiotic resistance profiles of E. coli in raw and RTE stuffed mussels. It did provide some useful data; however, the significance of this research is not described clearly, and the results obtained in this manuscript were not fully discussed. Therefore, it is not suitable to publish in Pathogens at the current state. There are some suggestions for consideration.

1.     Line 20, why “unauthorized harvesting and sales practices” was specially proposed? Did any samples isolate from that condition?

2.     Line 26, did “Antibacterial resistance” and “antibiotic resistance” represent the same meaning? If so, please unify.

     3. Line 55, why did these authors choose E. coli? Just because it's an indicator bacterium? Any other reasons? Please provide detailed instructions.

4. Lines 125 and 128, are there any updated versions of these detection methods?

5. Are there any differences between isolates from different months? Please provide detailed instructions.

Comments on the Quality of English Language

English editing required

Author Response

Comments and Suggestions for Authors from Reviewer #3

Dear Reviewer,

Thank you for your valuable feedback on our manuscript. We appreciate your thorough review and constructive comments, which will undoubtedly help improve the quality of our work.

Comments and Suggestions for Authors from Reviewer #3

This manuscript had a basic investigation on the prevalence, molecular characteristics, and antibiotic resistance profiles of E. coli in raw and RTE stuffed mussels.

  • It did provide some useful data; however, the significance of this research is not described clearly, and the results obtained in this manuscript were not fully discussed. Therefore, it is not suitable to publish in Pathogens at the current state. There are some suggestions for consideration.

Authors’ response: Thank you for your constructive feedback regarding the depth of the discussion in our manuscript. We acknowledge that the initial submission did not sufficiently explore the implications of our findings. In response to your comments, we have thoroughly revised and expanded the Discussion section.

The enhanced discussion now more comprehensively addresses:

  • The potential mechanisms underlying the observed patterns of coli contamination.
  • The implications of these findings for public health, especially in contexts where food safety regulations may be less stringent.
  • Comparative analysis with existing literature to place our results within the broader framework of current research on microbial contamination in aquatic food sources.
  • Detailed consideration of the preventive measures and regulatory strategies that could mitigate the risks associated with coli contamination in shellfish.
  • We believe that these enhancements have substantially strengthened the manuscript by providing a deeper insight into the significance of our studys' results and their relevance to both public health policy and future research directions.

We appreciate your guidance in enhancing the clarity and focus of our manuscript.

Comments and Suggestions for Authors from Reviewer #3

  1. Line 20, why “unauthorized harvesting and sales practices” was specially proposed? Did any samples isolate from that condition?

Authors’ response: Thank you for your insightful comment regarding the specific mention of "unauthorized harvesting and sales practices" in our manuscript. Upon reflection, we acknowledge that this phrasing may have inadvertently suggested that our study samples were obtained from unauthorized sources, which is not the case. All E. coli isolates analyzed in our study were indeed sourced from companies applied the Hazard Analysis and Critical Control Points (HACCP) and Good Manufacturing Practices (GMP) systems in production (2. Materials and Methods; 2.1. Sample Collection). These establishments adhere to rigorous standards for harvesting and processing mussels, ensuring adherence to all applicable food safety regulations.

The reference to "unauthorized harvesting and sales practices" was intended to underscore a broader public health concern applicable globally, particularly in less regulated markets where such practices might prevail, potentially increasing the risk of foodborne illnesses. However, we realize that this statement could be misleading in the context of our studys' data.

For the Abstract section of your manuscript, where you want to begin with a clear statement of the scope and implications of your study, here's how you can structure your opening sentence:

"Our study focused exclusively on analyzing E. coli contamination in fresh raw mussels and ready-to-eat (RTE) stuffed mussels obtained from authorized and regulated facilities; however, it is critical to recognize that such contamination represents a significant public health threat in regions where unauthorized harvesting and sales practices are prevalent."

Comments and Suggestions for Authors from Reviewer #3

  1. Line 26, did “Antibacterial resistance” and “antibiotic resistance” represent the same meaning? If so, please unify.

Authors’ response: We appreciate the reviewers' feedback on the terminology used in our manuscript. We have reviewed the entire manuscript and have replaced the term "antibiotic" with "antibacterial" in all necessary instances to maintain consistency.

Thank you for your thorough review and valuable suggestions.

Comments and Suggestions for Authors from Reviewer #3

  1. Line 55, why did these authors choose coli? Just because it's an indicator bacterium? Any other reasons? Please provide detailed instructions.

Authors’ response: Thank you for your insightful question regarding the selection of Escherichia coli (E. coli) in our study. We appreciate the opportunity to clarify our rationale.

The aim of our project, which includes this study, is to investigate STEC serotypes that have never been investigated in our country and are not known to occur in mussels anywhere in the world. Although E. coli is known as an indicator microorganism, as the reviewer stated, serotypes such as STEC, EHEC, EPEC are known as important human pathogens other than the known indicator E. coli. There are not many studies on the prevalence of serotypes such as STEC, EHEC, EPEC and ETEC in mussels. Therefore, in this study, we aimed to serotype our E. coli isolates according to the characteristic of carrying the stx and eae genes according to the ISO 16049 method, and then to evaluate all isolates carrying or not carrying stx and eae according to the public health concern perspective. This statement, which we made in response to the reviewers' valuable comments, has been added to the line 58-68 of the Introduction section. In addition, the purpose of the study in the line 92-96 of the Introduction section has been revised in line with the reviewers' suggestions.

Information and references on the importance of the study have been added.

Comments and Suggestions for Authors from Reviewer #3

  1. Lines 125 and 128, are there any updated versions of these detection methods?

Authors’ response: We appreciate the reviewer's attention to detail and their valuable question regarding the detection methods mentioned in lines 125 and 128.

Clermont et al. (2000) identified three phylogroups in the phylogenetic classification of E. coli isolates. In 2013, Clermont et al. expanded this classification to include five phylogroups using a different method. Our study also focused on detecting these five phylogroups and incorporated advanced techniques, such as genome-based classification, following the suggestion of Reviewer #2. The cost of genome analysis per isolate is estimated to be $5,180, considering the average cost is $70. This cost would be approximately 165,000 TL in our country at the current exchange rate. While this may require a large budget, we plan to implement genome-based classification in our next study, as recommended by the reviewer. In addition, this situation is also presented as a recommendation in the discussion (line 314) and conclusion section (line 411-413).

Comments and Suggestions for Authors from Reviewer #3

  1. Are there any differences between isolates from different months? Please provide detailed instructions.

Authors’ response: We appreciate the reviewer's insightful comment regarding the differences between isolates from different months. In response, we have revised the Results and Discussion sections to provide more detailed analysis of all findings. Specifically, the differences between the isolates across the months have been elaborated upon in lines 197-203, 222-223, 231-233, and Figure 1 (In the figure, the monthly differences are more clearly visible than in the Table) (in Result section); Lines 299-301, 314-318, 335-345 (in Discussion section).

Comments and Suggestions for Authors from Reviewer #3

  1. Comments on the Quality of English Language

English editing required

Authors’ response: Thank you very much for your valuable suggestions. We have submitted our manuscript for editing after receiving revision suggestions. Our institution, Bursa Uludag University, provides proofreading services for articles revised in Q1 and Q2. Revisions, including title changes, can be seen in the article file. By providing English language support, we aim to improve the clarity of the manuscript and increase its citation potential. Authors have carefully reviewed the revised manuscript and have highlighted all changes in yellow for easy tracking. We believe that these revisions address your suggestions. Any further feedback is welcome.

We hope that these revisions address your concerns and improve the clarity and comprehensiveness of our manuscript. Thank you once again for your insightful comments.

Round 2

Reviewer 2 Report

Comments and Suggestions for Authors

In my opinion, the manuscript can be published in this form.